# Navigating Digital Geographies and Trauma Contexts: Conceptions of Online Communities and Experiences Among LGBTQ+ People During COVID-19

**DOI:** 10.3390/ijerph22030443

**Published:** 2025-03-17

**Authors:** Rachel M. Schmitz, Jennifer Tabler, Ruby Charak, Gabby Gomez, Reagan E. Cole, Joshua J. Schmitz

**Affiliations:** 1Department of Sociology, Oklahoma State University, Stillwater, OK 74078, USA; gabby.gomez@okstate.edu (G.G.); reacole@okstate.edu (R.E.C.); 2Sociology Program, University of Wyoming, Laramie, WY 82071, USA; jtabler@uwyo.edu; 3Department of Psychological Science, University of Texas Rio Grande Valley, Edinburg, TX 78539, USA; ruby.charak@utrgv.edu; 4Engineering Department, Embry-Riddle Aeronautical University, Daytona Beach, FL 32114, USA; schmij38@my.erau.edu

**Keywords:** LGBTQ+, online trauma, community, social technology, digital geographies

## Abstract

The coronavirus pandemic shaped challenges for marginalized groups. Specifically, lesbian, gay, bisexual, transgender, and/or queer (LGBTQ+) people experienced community-building constraints, notably in predominantly rural regions. People are also navigating digital geographies, or online social environments, in novel ways to develop virtual communities in the face of prejudice, discrimination, and potential trauma. Through a minority coping approach, the present study explored LGBTQ+ people’s experiences navigating the dynamics of digital geographies during the pandemic while residing in socially conservative, highly rural physical spaces where they may be exposed to vicarious trauma. Using qualitative semi-structured interviews, data were gathered from 43 LGBTQ+ identifying individuals between 19 and 59 years old (M/SD = 27.7/9.2) between October 2020 and January 2021. Nearly 14% identified as transgender, nonbinary, or queer individuals, 35% as bisexual individuals, and 21% as people of color including Hispanic/Latina/o. Thematic analysis of the narratives described participants’ exposures to online discrimination and stigmatization of minority groups (racial and/or sexual/gender minority groups) during the COVID-19 pandemic, institutional constraints to identity expression, utilizing social technologies to manage their identities, and negotiating digital strategies to promote social ties. Findings emphasize improving marginalized people’s experiences with digital geographies through identity affirmation and community relationship-building to offset potentially traumatic experiences. Furthermore, service providers can utilize the findings to tailor effective virtual LGBTQ+ community programming to support underserved, marginalized populations.

## 1. Introduction

The COVID-19 (coronavirus) global pandemic shaped health challenges among many marginalized groups, including people in poverty, people of color, and people who identify as gender and/or sexual minorities [1]. Specifically, mental wellbeing challenges heightened as a result of COVID-19, with mental health disparities growing among marginalized groups, such as heightened depression and anxiety [2]. Mental and emotional health is tied to community connectedness, particularly for socially marginalized groups that rely on fellow community members for social support, such as lesbian, gay, bisexual, transgender, and/or queer (LGBTQ+) people [3]. In particular, rural US regions characterized by socially conservative policies can drive community disconnection among people as they are often under-resourced and rife with exposure to vicarious trauma through prejudice and discrimination [4]. With such variable access to social support in the United States, it is critical to examine the nuances of how the pandemic shaped wellbeing and community connectedness for LGBTQ+ populations in the face of adversity.

As pandemic-related topics became highly politically controversial surrounding vaccines and mitigation efforts (i.e., mask mandates), the social stress of the pandemic concurrently escalated [5], especially among social groups whose lives are historically politicized, such as LGBTQ+ people [6]. In turn, feelings of social isolation and elevated online activity in work, education, and social media all intersected to impact wellbeing in distinctive ways as people navigated the pandemic’s enduring societal turmoil. More and more people began utilizing internet-based communication technologies (e.g., Zoom, Teams, Skype, Facetime) to manage both personal vulnerability (e.g., high-risk health categories) and structural restrictions (e.g., remote work requirements/options) [7]. Due to this decrease in physical social interaction, many people struggled to cope with feelings of social detachment, isolation, and loneliness, which often worsened wellbeing in rural areas where strong kinship ties can be critical to mental health [8,9]. When devices and reliable internet are accessible, people may turn to technology for social connection (i.e., “e-support”) [10]. Indeed, LGBTQ+ people have long utilized digital geographies to connect with peers and secure information when offline resources are scarce [11]. Digital geographies are diverse online environments where people go to socially interact, with varying individuals and groups exhibiting unique virtual mobility as they produce, access, and navigate online spaces [12]. These digital environments can include social media platforms, emails, and teleconferencing platforms that are all accessed through electronic devices such as computers and smartphones. Increased internet and device usage, however, can also be a source of stress and even potential trauma when access is unequally distributed or it shapes adverse experiences like stigma and discrimination [13].

People from marginalized social groups, such as LGBTQ+ people, were vulnerable to adverse pandemic-related outcomes due to deeply embedded social inequalities [14] and the minority stress stemming from anti-LGBTQ+ sentiment tied to both gender and sexual discrimination [15,16,17]. In general, LGBTQ+ people face elevated mental health inequalities due to minority stress processes [18], including disproportionately declining wellbeing in the pandemic compared to non-LGBTQ+ people [19]. Physical distancing inhibited community relationship-building among LGBTQ+ people, with online communities helping to mediate this relationship [20,21]. While online resources can serve as a group-level resource for LGBTQ+ people engaging in minority coping to develop strengths to resist minority stress [22], digital geographies are also socially constructed to reinforce inequalities which can shape marginalization [23]. Through a minority coping approach to technology, this study explored the following research question drawing from in-depth interviews with 43 LGBTQ+ people living in rural and socially conservative U.S. states: How did LGBTQ+ people experience and navigate pandemic-driven digital geographies surrounding their marginalized identities and sense of community? This study can inform scholars and practitioners on the complexities of minority coping in response to minority stress, technology use, exposures to vicarious trauma, and community connections among LGBTQ+ people.

### 1.1. Background

#### 1.1.1. Pandemic-Induced Social Inequalities

The COVID-19 pandemic shaped increased inequalities for people in the United States, with an increase in rates of mental health challenges, financial concerns, and social disconnection [24]. Fear around COVID-19 coincided with the widespread political and ideological conflict surrounding pandemic topics such as mask-wearing, social distancing, and vaccination [25]. Pandemic politics were notably divisive and controversial across the United States, leaving many underserved communities in a state of community fragmentation and disconnection [26]. In these ways, societal stressors related to the physical dangers of the pandemic transformed into significant social discord that spotlighted preexisting societal inequalities.

Specifically, LGBTQ+ people were especially vulnerable to pandemic-related stressors based on their derogated social status [27] that can shape minority stress [16,17]. Further, LGBTQ+ people were more vulnerable to COVID-related health outcomes based on their higher prevalence of various health challenges and social marginalization [28]. LGBTQ+ people’s mental health was harmed by pandemic mitigation efforts (i.e., social distancing, quarantine, shelter-in-place), including heightened stress [29]. Additionally, LGBTQ+ adults reported some of the highest rates of pandemic-related worry and stress (e.g., fear of infection, concern for the safety of friends and family), which could be tied to their reliance on constructed kinship for support [30,31]. Considering structural wellbeing, LGBTQ+-identifying households also reported elevated rates of job loss, food insecurity, and inability to access needed health care [4]. Overall, the preexisting social inequalities worsened by the pandemic disproportionately impacted LGBTQ+ populations in fundamental ways, with more understanding needed of LGBTQ+ people’s experiences navigating digital geographies during this time while living in socially conservative regions.

#### 1.1.2. Community-Building Among LGBTQ+ Populations

Drawing on social supports can be key in buffering sexual and gender identity-related marginalization and minority stress for LGBTQ+ people [32]. Historically, LGBTQ+ people may establish supports in wider queer communities who often share similar trauma histories and through their own creation of “personal communities” [33]. Pandemic-related processes such as social distancing and sheltering-in-place were especially isolating for LGBTQ+ individuals by limiting social support opportunities that can mitigate minority stress [34]. In one study, Australian gay and bisexual men described a loss of sexual identity-based social ties as well as a diminished sense of belonging to queer communities [20]. Some LGBTQ+ people drew collective strength from well-established community and cultural values during the pandemic [35]. The complex impacts of the pandemic reshaping community-building and relationships among LGBTQ+ populations warrant further investigation into LGBTQ+ people’s diverse responses and coping strategies, particularly in the realm of navigating digital geographies.

#### 1.1.3. Internet and Social Technology Usage During the Pandemic

Overwhelmingly, people’s use of internet-related technologies across the world increased since the onset of the COVID-19 pandemic, with people adapting to virtual, remote work, education, and telehealth configurations [36]. While such technology use can certainly serve as a societal coping tool to stay socially engaged and informed during the pandemic [37], it can also drive stressors (i.e., “social media fatigue”) as people sift through disinformation and political divisiveness in their digital geographies [38]. Identity-based violence can result in potentially traumatic outcomes for marginalized groups [39], so stressful experiences of digital geographies may also drive exposure to vicarious trauma for marginalized groups when they perceive stigmatizing and anxiety-inducing messaging online [40]. The nuances of expansive internet technology use during COVID-19 are also underscored by sweeping social inequalities in terms of technological access, experiences, and opportunities that beg further sociological inquiry [41].

In addition to highlighting structural disparities, the pandemic also revealed challenges in how people have adapted to COVID-mediated internet usage, with socially marginalized populations more at-risk for harm through unequal technology experiences [42]. While COVID-19 created unique challenges for LGBTQ+ people, many also found safe and inclusive support through online outlets [19]. Attachment to supportive online queer personalities helped to offset mental health harms like depression and loneliness for LGBTQ+ young adults, particularly those with tenuous family support [43]. Sexual minority men reported utilizing social technologies while physical distancing, even though distancing shaped mental distress [21]. Digital communication platforms also enabled queer couples to develop intimacy when COVID-19 lockdowns physically separated partners [44]. LGBTQ+ people have often widely utilized the power of digital geographies and social media platforms to create communities and drive advocacy [45], especially following shared collective traumatic experiences that threaten LGBTQ+ people’s sense of public safety [46]. However, LGBTQ+ people reported disproportionately reduced rates of internet access during the pandemic, showing technology-based community-building is not equitably available for all [4]. The concurrent rise in social inequalities and internet use during the pandemic warrants the present study’s multifaceted examination of how LGBTQ+ people navigated digital geographies to describe their experiences of COVID-19 at the intersection of social marginalization, community relationship-building, and digital engagement.

### 1.2. Theoretical Background

#### 1.2.1. Digital Geographies and Identity Development Online

The use of online social spaces and communities, or digital geographies, has become increasingly more prevalent in people’s lives with the growth of the internet and virtual interaction mediums. In particular, utilizing digital geographies for social support and community can be especially critical for marginalized people living in predominantly rural, socially conservative regions where in-person resources are limited. LGBTQ+ people, notably young adults in rural regions, have accessed and reshaped digital geographies to develop their identities in inclusive virtual settings when social support in their physical geography can be precarious due to anti-LGBTQ+ ideologies [47]. In navigating marginalization, LGBTQ+ people often strategically shape their coming out and identity expression processes [48], and shifting to social interactions in digital spaces necessitates unique impression management strategies that can be challenging to negotiate. LGBTQ+ people often face the significant labor of modifying their identity performances (i.e., “digital personhood”) in diverse digital social networking geographies contingent on perceptions of acceptance and bound by personal understandings of social expectations in various platforms [49]. The importance of exploring LGBTQ+ people’s impression management experiences within the pandemic context is paramount as interactions within major social environments became largely virtual, including digital offices and classrooms.

#### 1.2.2. Minority Coping Framework

Although LGBTQ+ people face heightened societal marginalization, queer populations are also adept at establishing strengths and assets to resist structural oppression and exposure to potential trauma sources [50]. Complementary to minority stress, the minority coping framework is useful for exploring diverse ways that LGBTQ+ people navigate, manage, and challenge prejudice and discrimination through group-level supports and relationships [16,22]. For example, minority coping among LGBTQ+ people, including community support systems, self-care practices, and educational empowerment through advocacy, can build resilience in helping LGBTQ+ people overcome minority stressors tied to marginalization [22]. This framework further emphasizes the importance of how online communities within digital geographies can facilitate individuals in capacity-building and strengthen individual wellbeing. While digital geographies can promote unique opportunities for community-building, internet-based, technology-mediated social interactions are bound by broader hierarchies of social power and structure [23], leading to variations in experiences of minority coping or community strengths among LGBTQ+ people [22]. We use the minority coping framework to explore LGBTQ+ people’s complex experiences navigating digital geographies amidst a variety of minority stressors online and engaging in minority coping to build resilience during COVID-19 (Figure 1).

## 2. Materials and Methods

### 2.1. Procedure and Participants

Data for the present study are based on a larger mixed methods project described in full elsewhere [29]. Data collection occurred between October 2020 and January 2021. Drawing from the research team’s institutional affiliations and social networks, both purposive and convenience sampling strategies were employed that targeted university employee and student listservs in three highly rural and socially conservative states—Oklahoma, Wyoming, and Texas. Therefore, our recruitment strategy targeted and oversampled for rural-residing residents with heightened exposure to socially conservative political climates. Additionally, to oversample for LGBTQ+ respondents, flyers of the study were posted on local LGBTQ+ social organizations’ social media pages. Interested participants were invited to complete a self-administered online survey via secure survey link (Qualtrics™), survey respondents who identified as LGBTQ (*n* = 129) were asked to indicate if they were interested in participating in a follow-up interview. A member of the research team contacted all participants who indicated interest via email to schedule the interview. The current study draws only from the qualitative semi-structured interviews conducted with 43 LGBTQ+-identifying people who were recruited from the larger sample of quantitative survey participants.

### 2.2. Data Collection

Authors RMS and RC conducted all interviews in English. Study participants completed one audio-recorded, in-depth interview lasting approximately one hour and a short demographic questionnaire. All interviews were conducted remotely via Zoom. Study procedures were explained to participants and verbal informed consent was obtained prior to the interview after participants asked any questions. Participants received a USD 20 gift card in exchange for their time. All respondents were asked the same series of 20 open-ended questions surrounding their pandemic experiences (see interview guide in Appendix A) in a semi-structured format to ensure participants could guide the interview flow. A constructivist approach guided this study’s design and development including recruitment materials, interview questions, and subsequent coding strategies, to ensure inclusive, person-centered interpretations [51]. Specifically, the interview guide questions were developed with a focus on emergent empirical findings surrounding pandemic experiences and through the lens of our guiding theoretical frameworks emphasizing digital geographies and minority stress and coping among LGBTQ+ people. To ensure participant wellbeing, the interviewers checked in regularly with participants throughout the interview on how they were doing and if they needed a break. Further, each participant was given a list of community-focused resources and the interview team designed a Crisis Protocol to utilize in case a participant exhibited extreme duress, which included providing a mental health hotline and devising a safety plan together (this protocol was never deemed necessary during data collection). Participants self-assigned pseudonyms to promote confidentiality. The university institutional review board (IRB-20-419) approved this study.

### 2.3. Data Analysis

All interview audio recordings were transcribed and uploaded into MAXQDA 2020 for analysis. Authors RS and GG collaborated as the primary coders to conduct multiple rounds of coding following thematic analysis to promote rigor and validity and an interpretive framework [52]. We began with open coding to identify general patterns of LGBTQ+ people’s pandemic-related experiences and impacts on their wellbeing. Next, the coders deployed axial coding to establish connections among the codes, such as connecting codes for varied COVID-related perceptions (i.e., mixed and changing over time, potential to learn from it) and political understandings (i.e., political divisiveness, fear of social unrest). Finally, the coders used selective coding to delineate overarching categories, or domains, which comprised our final themes. The primary coders reached coding agreement through multiple collaborative discussions to discuss the codebook’s development. Authors JT, RC, and JS provided supplementary analytic support by reviewing the final codebook and emerging themes and offering feedback. In the rare instances of coding disagreement, we conducted additional analyses until consensus, such as through code modification or disconfirming evidence identification. Following each round and stage of coding, we met to discuss validating feedback, which involved iterative memoing of coding decisions and collaborative discussions of theme construction to enhance the study’s trustworthiness [53]. Data saturation was reached when coding concluded based on the replication of codes and themes across transcripts and the lack of new insights emerging [54]. Data, methods, and materials are available upon request by emailing the corresponding author.

## 3. Results

The final analytic qualitative subsample included 43 LGBTQ+-identified people aged 18+, with an average age of 28 years old. Around 44% (*n* = 19) of the sample identified as currently living in a rural community. Table 1 lists the participants’ self-identified sociodemographic details. Firstly, participants highlighted their exposures to potential vicarious trauma that caused feelings of distress and sadness. Secondly, LGBTQ+ people described experiencing challenges with technology-mediated communication platforms, or digital geographies, organized through their institutional affiliations, such as education and the workplace, which constrained processes of virtual identity and community relationship-building. Thirdly, participants engaged with digital communication platforms in deliberate ways to effectively navigate their multiple marginalized identities in social interactions. Finally, participants also emphasized establishing innovative digital coping strategies through online community formation to contend with—and potentially offset—traumatic marginalization experiences.

### 3.1. Witnessing Discrimination and Stigmatization of Minority Groups Online

The pervasive stigmatization and marginalization of minority communities was magnified during the pandemic. Participants highlighted how online platforms became arenas where discriminatory behavior intensified, amplifying the psychological and emotional toll on both the targeted communities and digital observers. In discussing her concerns around safety and feelings of distress, Chrissie, a cisgender bisexual woman, stated:
It’s very concerning; those flags on cars, that just seem constant. And, I agree, growing in number… it makes me scared, to an extent, not so much for my safety, a little bit for my safety, anyone who knows me personally will be like, ‘yeah, no, she’s definitely left’. If you just look at me, people aren’t going to know that I’m bisexual or I’m like, really feminist and really pro-choice, pro-black… so I feel like there’s a level of passing for myself. But that doesn’t give me any comfort.

Similarly, Bailey adeptly employed a prior analogy of her friend getting uninvited from being the maid of honor at a wedding, an act that was “very homophobic and sexist”. The fact it came from within the LGBTQ+ community, validated the existence of [online] discrimination and stigmatization experiences of minority groups that Bailey described as “right now, it is very clear that the true values of people are coming up”. In agreement, Benjamin stated that during the pandemic it was clear how people do not care about anyone “who do not look like them” and are willing to go to the extent of typing derogatory statements online with no regards to an individual’s physical and psychological safety and feelings.

### 3.2. Managing Institutional Constraints to Virtual Identity Negotiations

Many participants shared the common sentiment that institutional transitions towards remote/virtual modalities created a number of challenges they struggled to navigate, especially in terms of virtual identity and community-building endeavors. In discussing the difficulties of virtual learning modalities compared to in-person, Cassandra, a cisgender bisexual woman, lamented how the loss of in-person interactions with classmates could potentially harm their acceptance of her and lead to potential prejudice:
I count on a lot of face-to-face interaction for them to see me as I really am. I am a very extroverted personable person and I think that most people, if they get to know you before they find out these big things that they might not agree with, that it’s almost easier for them to see you as a real person instead. It’s been weird cause it’s like, all they see of me is the like two or three hours a week we have. And if I say something related to my sexuality, that’s going to be one of the major bullet points they know about me. And I don’t want that to really color their opinion negatively of who I am as a person.

Echoing the challenges of institutional identity management and presentation of self, Laney, a cisgender bisexual woman, also shared that she adopted a strategy of “disclosing less” in virtual training formats “where these people haven’t gotten to know me as a person first…I want people to know me for who I am before they need to know anything about my orientation or identities”. In a variety of ways, as indicated by participants, managing how one’s LGBTQ+ identity is revealed via technology-mediated communication platforms was no easy task and often required significant labor, such as suppression of self and strategic outness.

Similarly to Cassandra working to manage her identity in terms of disclosure with colleagues via video conferencing, some participants also recounted the challenges of balancing their presentations of self that were highly visible on digital interactions and perceived as under heavy scrutiny. Chelsea, a pansexual queer person, described feeling that virtual interactions with school colleagues compelled her to closet her identity in a sense:
I definitely feel the increase of stress, regardless if it’s the pandemic or with the election or with the school or whatever. Using technology kind of forces me to go towards the normal… like the status quo. I mean it’s easier to live in [city, state] as a straight white female that is more feminine than anything else, right? Like, I feel like I can’t be overly masculine in some stuff, like my clothes or my actions. Like, I feel like I can’t even express my identity, because it’s easier to see these things, so I put on sparkly earrings and do my hair and just fit in as normal because that’s one less stress I have to deal with.

Managing one’s LGBTQ+ identity in the pandemic via technology platforms, therefore, did not involve only verbal strategies, but also prompted participants to carefully consider how their physical expressions of identity might be interpreted in stigmatizing ways. As Chelsea stated, suppression of one’s expression of self, particularly through conforming to “the normal” in more conservative, predominantly rural regions that did not align with their true gendered or sexual self, can create challenges in managing one’s identity within online institutional settings.

Particularly at the outset of the pandemic, LGBTQ+ participants recounted stressors stemming from institutional lags in supporting gender and sexual identity expansiveness in identity markers, such as pronouns and people’s names. Layla, a transgender woman, described initial aggravation with institutional deficits and then an eventual adaptation to this challenge navigating digital geographies: “At first I got pretty annoyed whenever it’s like, “Oh, my email is showing my nonpreferred name. And classmates were calling me my nonpreferred name that was ticking me off a lot. But nowadays it’s just like the wrong email address got sent incorrectly. I’m not really coping, just like rolling with the punches.” For LGBTQ+ people like Layla, it can feel futile to continually fight institutional failures in acknowledging their gender pronouns. Narratives such as Layla’s directly highlight institutional gaps in technological capabilities of allowing users to self-identify using markers and names that most closely align with their authentic selves. Despite admitting she was not “really coping”, but simply trying to adapt by “rolling with the punches”, it is clear that this issue recurred for Layla and was a source of repeated institutional stress as she attempted to navigate her educational interactions while being deadnamed.

### 3.3. Navigating Marginalization in Digital Communication Interactions

In addition to technology shaping new and challenging institutional interactional issues, the pandemic also created challenges in the form of various social technology geographies, including social media and online communication platforms (e.g., Zoom), in which LGBTQ+ people struggled to navigate marginalization. Linking his struggles with both professional interactions as well as digital modes of social interaction, Luca, a cisgender gay man, described the struggle of being “taken seriously” on video call interviews intersecting with his gender, sexuality, and ethnicity:
Is my voice deep enough… I don’t want to be seen as the weak gay man. And so, I might lower my voice a little bit more. And I’ll try to sound white. So I’m very careful with some of the words I use and how my accent comes across, because I don’t want to be seen as a Latino gay male. I want to be seen as just a male.

Luca’s experience is especially striking in how he struggled to navigate impression management across his multiple marginalized identities, which underscores the effort and labor he applied to shifting to video calls with a variety of professional colleagues. In learning to navigate the digital geography of relying on video conferencing as a primary mode of social interaction, especially in socially conservative regions with fewer opportunities for in-person social engagement, people from multiple marginalized groups endeavored to remain cognizant of how their identities might be stigmatized. While participants like Luca discussed strategic identity management as a way of avoiding prejudice and discrimination on video calls, this type of identity concealment can also serve as a source of traumatic stress in and of itself when marginalized people feel they must codeswitch to make the right impression.

In a related but distinctive way, LGBTQ+ participants also discussed the digital communication challenges imposed by the pandemic in the pursuit of intimate partnerships and interactions. Max, a gay cisgender man explained that solely being able to rely on virtual interactions when meeting potential dating partners induced feelings of worry and anxiety as the digital context was not conducive to him expressing his holistic, genuine self:
It stresses me out that I have to try to make a personal connection with someone I’m only texting, where before I could say “Hey, let’s go on a date. Let’s meet in person”. I feel like I can be more of myself in person than I can be online through social media or texting. It made it easy to separate like potential boyfriends in person, because you could see if that connection is there or not. I’m also afraid that I’m misrepresenting myself with someone, like I’m trying to be cooler. Cause you have time to sit and think about your reactions or what you could text back. To me, that’s not really being genuine.

For LGBTQ+ people like Max, impression management in the dating scene took on unique challenges in the pandemic when it was difficult to make deeper intimate connections that seemed authentic.

Some participants described institutional attempts, such as employers, to help people socially connect, or as Shelly, a cisgender pansexual woman put it, “marry technology with finding a network that connects to your identity.” Greg, a gay cisgender man, shared his frustration with “stymied” attempts at LGBTQ+ community-building at his workplace (located in a socially conservative state) in the midst of pandemic-based restrictions on social gatherings:
Being unable to meet in person and everybody being so disconnected that they don’t want to meet on Zoom. And struggling to maintain those connections. We started this LGBTQ+ employee network thing this past year and have been kind of stymied with the technology and people not wanting to engage on that platform. So I would say that is causing the stress, because as much as it’s aiming to meet the needs of the constituents, it’s failing to bring people together in a meaningful way.

While Shelly acknowledged that these networks had the potential for meaningful social connections, her membership in both an LGBTQ+ and women’s network was not fruitful because “I’m so tired of looking at my computer all the time, and I’m really stressed out… I don’t feel like I can be my full self. It is very hard to open the Zoom link and engage”. So, on one hand, community organizers are struggling to create digital opportunities for engagement, but garnering participation still proved to be a major challenge. In these examples, Zoom fatigue proved to be a real phenomenon that can create obstacles to meaningful virtual interactions when technology-mediated social platforms may be the only option for interpersonal connections and they are also paired with an uptick in technology use throughout people’s lives.

Socially deployed uses of technology in the pandemic could also shape feelings of marginalization for LGBTQ+ people, even in indirect ways. Bethany, for example, a cisgender bisexual woman, found the internet to be a “really powerful tool to connect with other people like me, especially living in rural [name of the state]”, but also struggled to maintain identity confidentiality in her household:
I’m not out to my mother and like, my gay life lives on my computer. In some ways, it’s almost a liability. We watch Netflix all the time together, but she’ll just, you know, use my profile on the TV and it’ll have, like, I was watching *Blue is the Warmest Color*, which is basically lesbian porn. And it was going to pop up, like continue watching. So I just grabbed the remote and I was like, “Oh, let me just log onto your profile”.

Even when not actively engaging in social media, simply consuming it by scrolling could be distressing for LGBTQ+ people when it increased in the pandemic and primed them for considering their own identities and life trajectories. One case is Renee, a pansexual cisgender woman, who described self-critiquing her life choices based on the heteronormative patterns she perceived in her socially conservative-leaning social networks:
Well I think there’s like the stress of, specifically with social media, like seeing a lot of people from my small-town high school or some of my undergraduate friends had gotten married and had kids. I guess there’s like that stress and that pressure trying to be independent, but seeing people on the outside doing these more traditional life path type things.

Processes of internalizing stigma and perceptions of dominant cisheteronormative social norms perceived through social media outlets can be uniquely challenging for LGBTQ+ people in navigating digital geographies and communication. In particular, LGBTQ+ people must adapt to virtual social interactions through the lens of marginalization and are often forced to reflect on their own identities and how they might contrast with dominant societal expectations.

### 3.4. Establishing Digital Coping Strategies for Community Building

LGBTQ+ participants were also actively using digital apps and technologies to promote their wellbeing and sense of community during the pandemic, even when technology could simultaneously be very challenging to navigate. Digital geographies like social media were often used to establish and maintain social support relationships for people in socially conservative regions with limited opportunities for safe in-person socializing, like in the case of Erika, a nonbinary bisexual person who appreciated Facebook updates from family members:
During the quarantine, a lot of my family members were posting, and it was just fun to see them doing okay. It was really helpful towards my mental health because it made my stress level go down, that way I could know that they’re doing okay.

Other participants distinguished the benefits and pitfalls of certain social media compared to others, and strategically balanced their exposure knowing how it would impact their psychological wellbeing. Allie, for example, a cisgender pansexual woman, preferred Instagram and being able to “post pictures here and there and just putting things out there that are funny or uplifting or speak to me… it’s a nice break from Facebook where there can be a lot of vitriol and so many triggering things”. So while social media and other social technologies can create marginalizing experiences in certain situations, curated uses of digital spaces can also be beneficial to LGBTQ+ people by allowing for strategic community building.

In various ways, participants also described technology-assisted communication as improving their relationships overall by providing novel and engaging methods of social interaction. Chrissie explained this dynamic in the following way:
I’ve gotten better at talking to my friends. We might not hang out in person as much, but just texting. So I do feel more connected and I feel like the idea of an online relationship has been normalized more throughout COVID.

Similarly, Greg, a cisgender gay man, found social media like Snapchat to provide an extra layer of visual connection with someone that he would not receive with simply a phone call or text:
I really like to Snapchat with both friends who I typically communicate with by Snapchat, but also people who aren’t able to get together since we live kind of spread out. It’s nice to be able to see their face versus a text message. Even if it’s not a video, just seeing somebody’s face in a Snapchat is, is really therapeutic.

These examples provided by LGBTQ+ participants show the promising potential of social technologies for positive social connection and community formation, especially during times of social instability when in-person interactions may not be feasible. In particular, meaningful online social connections can potentially be more “normalized” within COVID-19 and beyond, which could open up the accessibility of social supports for more LGBTQ+ people, like those living in regions with fewer opportunities to engage in-person.

For some participants, the internet and virtual technologies had long provided opportunities for positive social connection that was not as easily accessed in physical interactions, even before the pandemic. Cora, an asexual cisgender woman, explained her fruitful engagement with the internet in the process of identity development and establishing social support in predominantly rural, socially conservative community and upbringing:
It feels like, especially when I was younger and still in high school, the one place I could be free. I didn’t have to worry about second guessing myself because it’s like, okay, if I said this, it could get back to someone and I would have to commit to it. When I might’ve just been experimenting or talking. And I still feel that way even today with the internet, it’s still like, okay, I can just go on the internet, look up ACE stuff and feel really great.

The narrative of Cora emphasizing the importance of the internet in her personal sexual identity development underscores the possibility of digital social geographies in supporting marginalized people when in-person community interactions can be stigmatizing. Relatedly, Kaley, a sexually fluid cisgender woman, found growing technology use in the pandemic to have “alleviated some level of stress because I get social anxiety sometimes, so the ability to have a little bit of distance helps me not feel as stressed”. While being homebound and having more time to be online, Kaley also reported utilizing social media, like connecting with queer content creators, which “gave me the nostalgia and the feeling of still being who I was before engaging in a heterosexual relationship” as well as helping her “feel less isolated than I would have without YouTube”. The fact that many LGBTQ+ people continually recognized the beneficial utility of the internet for their wellbeing, social connections, and community-mastery despite the potential technology-related stress induced by the pandemic illustrates how virtual communities can still be safe havens for people navigating trauma histories and marginalization.

## 4. Discussion

Utilizing a minority coping approach [16,22], study findings illustrate the complex dynamics and processes associated with how LGBTQ+ people navigated diverse digital geographies during the COVID-19 pandemic. Major themes centered on how LGBTQ+ participants dealt with increased engagement with digital social contexts and interactions and how this shaped their perceptions of marginalization including vicarious exposure to marginalizations, identity development, and community building in socially conservative regions. Firstly, participants detailed their exposures to marginalizations as experienced by others, which validated the existence of ongoing online stigmatizations during the pandemic. Secondly, participants shared how managing institutional constraints within digital geographical contexts, like videoconferencing, created challenges surrounding impression management and social relationship formation. Thirdly, social technologies like social media and personal digital interactions for endeavors like job interviewing, dating, community relationship-building, and entertainment created challenges for LGBTQ+ people as they navigated not only individual impressions but also forming meaningful social relationships within digital geographies. Finally, in navigating general pandemic stress, participants were also actively devising innovative strategies to utilize digital geographies in productive, community-building ways. Beyond a dominant minority stress lens, our study’s findings contribute to empirical knowledge of marginalized social groups’ nuanced experiences with technology and virtual worlds during the pandemic. We also expand upon theoretical understandings of how digital geographies can create both challenges and opportunities for novel community connections for LGBTQ+ people living in socially conservative regions that are often more hostile to queer identities. The findings from the early COVID-19 pandemic era suggest long-term implications for LGBTQ+ individuals in the post-pandemic world as the increased reliance on digital spaces has reshaped how LGBTQ+ people navigate identity expression, community building, and activism, creating both opportunities for resilience and challenges like digital burnout and vicarious trauma from online marginalization.

Many participants shared that while they struggled with these LGBTQ+ identity-based struggles (detailed below), it was distressing and traumatic to witness the online marginalization and stigmatization of minoritized groups with visible identities (e.g., race) during the pandemic. Although such exposures were indirect, participants reported feelings of helplessness and distress in addition to concerns regarding their own safety in social environments that targeted individuals with minority identities. This finding underscores interconnectedness between individual and collective experiences, highlighting the broader implication of identity-based discriminatory experiences on the psychological and safety-related concerns of participants with any minority status. Furthermore, clinicians can draw from these findings to tailor trauma profiles of clients to be inclusive of vicarious trauma that can occur in digital geographies among marginalized groups [39].

The majority of participants discussed the LGBTQ+ identity-based struggles they encountered when attempting to navigate the sudden, heightened reliance on technology at the onset of pandemic closures/shutdowns for institutional demands, such as education and employment. Much research has examined how institutional contexts such as college environments and in-person workplaces can influence experiences of prejudice and discrimination for queer people [55,56]. In our study, LGBTQ+ participants emphasized the institutional constraints of identity expression within the digital geographies of workplace and educational contexts. These findings point to the need to consider institutional sources of marginalization for queer people when mediated through the internet when this is one of the few, if not only, accessible forms of social interaction. Institutions, therefore, should be particularly mindful of their online social settings to ensure inclusive engagement and respectful interactions. Participants’ experiences with professional impression management also map onto processes of emotional work enacted by multiple marginalized people to manage and potentially offset perceptions of prejudice and discrimination when interacting in structurally constrained environments [57].

Intertwining with their challenges in managing institution-based technological dynamics surrounding their identities, LGBTQ+ people also faced unique challenges in navigating their identities in a variety of social interactions. Coming out for queer people is an ongoing, dynamic process they must strategically manage when traversing different interactions, relationships, and environments, including regions that may not be as accepting or structurally supportive [48]. The present study’s findings expand upon strategic outness and identity management to also consider how technology-mediated forms of communication create distinctive complexities LGBTQ+ must deal with when exerting agency over their identity in social interactions. Our findings also extend Erving Goffman’s (1978) sociological framework of the presentation of self to people’s contemporary virtual lives online and how marginalized individuals may “self-edit” to promote positive perceptions [58,59]. Strategic identity management, performed online by LGBTQ+ participants, augments knowledge around how people perform a type of “respectability politics” within digital social interactions (i.e., self-censoring, conformity) so as to minimize the potential for stigma [60]. Being closeted can therefore create “sociopolitical trauma” for LGBTQ+ people when they feel shamed into oppressive silence as a survival strategy [61]. In these ways, the internet and virtual social contexts can create potentially traumatic “queer closets” for LGBTQ+ people when sexuality and gender-normative expectations persist as a reflection of wider dominant cisgender and heteronormative hierarchies [62].

Despite dealing with a multitude of marginalization stemming from social technology use in the pandemic, LGBTQ+ participants also resisted oppression by innovating digital-based coping strategies to build community relationships online. With digital geographies becoming increasingly prevalent in people’s lives, and notably increasing within the pandemic, virtual sources of social connection and communication can be crucial for marginalized people’s wellbeing. Participants’ narratives describing the effective use of online realms to cope with marginalization augments knowledge of the creative ways queer people utilize virtual worlds to carve out beneficial social relationships and resilience, especially when in-person resources may be scarce [4,19,47]. These findings can directly inform community-based support organizations deployment of effective online safe spaces for LGBTQ+ connection. Similarly, it is important to distinguish across types of social communication technologies and how they can differentially impact people’s wellbeing, such as interpersonal types (i.e., texting, video chatting) and mass-personal media (i.e., social media platforms) [63]. Our findings underscore the double-edged sword complexity of how online social interactions can simultaneously shape marginalization and support novel community-building efforts.

Balancing our study’s key contributions to understandings of pandemic-related technology stressors, it also points to areas of future research. Firstly, our study is constrained by a qualitative sample size of 43, so it is not generalizable to broader LGBTQ+ populations, and further quantitative analyses are needed to examine the macro impacts of increased internet usage during the pandemic for marginalized groups. As our sample is representative of the regions from which it was drawn, it is predominantly white. People of color are being disproportionately harmed by pandemic challenges, and inequalities can be exacerbated for people from multiple marginalized groups [64], and how they cope with intersectional minority stress [65,66]. Additional study is warranted to explore how people of multiple marginalized communities experience digital spaces during the pandemic through the lens of their intersecting identities, especially as white privilege permeates virtual contexts and racism informs digital queer intimacy hierarchies [67,68]. Finally, our study was conducted during the early stages of the pandemic and in recognizing COVID-19’s evolution, continued research is needed into how technology norms and practices are both impacting and being impacted by LGBTQ+ people. We are unlikely, therefore, to see a return to pre-pandemic digital engagement, thus highlighting the importance of critically reflecting on complex social dynamics and potential inequalities within digital geographies [23].

Our study also has distinctive implications for applications to policy and clinical domains. Anti-LGBTQ+ political legislation can have long-term harmful impacts on queer populations by creating hostile social environments that can be potentially traumatizing [69]. While progress has been made by institutional and structural technological outlets towards inclusion, such as some college learning management systems allowing pronoun identification [70], continued dedication to promoting gender and sexual equity and inclusion in digital institutional interactions is warranted by organizations [71]. Examples of inclusive and equitable organizational practices could include incorporation and normalization of pronouns into communications, allowing users to input their authentic names into organizational records, and promoting multiple modes of interaction contingent on comfort levels (e.g., anonymous surveys, audible and textual response options).

Additionally, we recommend providers serving LGBTQ+ people specifically screen clients for potential social technology/internet-mediated traumatic events to holistically tailor care [39]. Institutions serving particularly vulnerable, in-flux populations, like colleges and universities, should be especially mindful of how COVID-19 can hinder emerging adults’ life goals, along with mental health harms associated with a sense of “falling behind” [48]. Pandemic technology-based learning inequalities have persisted in higher education, with marginalized students (e.g., students of color, women, and first-generation) enduring elevated barriers such as access, contextual distractions, and loss of motivation, which underscores the need to develop more inclusive digital geographies [49].

## 5. Conclusions

Social inequalities persisted and worsened during the pandemic, with rural communities especially struggling with internet access and speed for online resources, showing how the internet is an unevenly distributed “essential public service” [72]. While much of U.S. society is no longer mandating or practicing physical distancing in our post-pandemic world, which can help promote community relationship-building, pandemic dynamics highlighted the lack of resources and supports for LGBTQ+ people specifically. While we have moved into a “post-COVID” era of our social world, people’s engagement within digital geographies is not diminishing, but is instead growing at an accelerated rate. Scholars, practitioners, and community members would do well to take the lessons learned from digital community interactions during pandemic shutdowns and apply them to building more inclusive and supporting virtual worlds for all. Practically, this includes structural recognition of both the power and peril of digital geographies for marginalized groups and implementing organizational practices and policies that center people’s online wellbeing. LGBTQ+ people’s early pandemic experiences remain relevant in the post-pandemic era by highlighting the need for inclusive policies and practices within digital and institutional spaces to address systemic inequalities, ensure respectful interactions, and support mental health through fostering social connectedness. Acknowledging technology’s complexity in impacting LGBTQ+ people’s experiences of both marginalization and community building, policymakers, and service providers should continue to utilize virtual social resources and supports in innovative ways while also balancing online interactions with safe in-person opportunities when possible.

## Figures and Tables

**Figure 1 ijerph-22-00443-f001:**
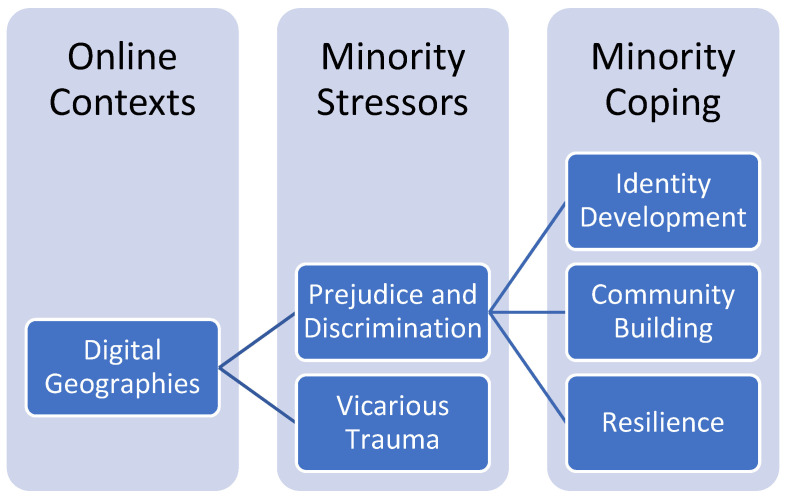
Minority coping in online contexts.

**Table 1 ijerph-22-00443-t001:** Descriptive statistics of the LGBTQ+ participants from the qualitative subsample.

Sociodemographic Variables (Qualitative Subsample; *n* = 43)	Sample Size/Percentage	*Mean/SD*
	*n* (%)	
*Age* (range = 19–59)		27.7/9.2
*Sexual identity*		
Lesbian	8 (18)	
Gay	7 (16)	
Bisexual	15 (35)	
Queer	3 (7)	
Pansexual	5 (12)	
Asexual	2 (5)	
Expansive sexuality/unlabeled	3 (7)	
*Gender identity*		
Cisgender women	27 (63)	
Cisgender men	10 (23)	
Nonbinary	3 (7)	
Transgender woman	1 (2)	
Queer	2 (5)	
*Race/ethnicity*		
White	34 (79)	
Bi or multiracial	3 (7)	
Latino/a or Hispanic	5 (12)	
Asian American	1 (2)	
*Regional identification*		
Rural	19 (44)	
Urban	9 (21)	
Suburban	15 (35)	
*Socioeconomic status*		
Working class	19 (44)	
Middle class	19 (44)	
Upper middle class	5 (12)	

Data are from primary data related to COVID-19 pandemic experiences, collected between October 2020 and January 2021.

## Data Availability

Data, methods, and materials are available upon request by emailing the corresponding author.

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
