# Peer review of "Navigating Digital Geographies and Trauma Contexts: Conceptions of Online Communities and Experiences Among LGBTQ+ People During COVID-19"

_ijerph, 2025, doi:10.3390/ijerph22030443_

Round 1
Reviewer 1 Report
Comments and Suggestions for Authors
This paper is focused on the important topic of the ways the digital world mediates mental health for LGBTQ+ people. The paper centres on the Covid-19 pandemic in particular but the analysis presented here has, as the authors argue, clear relevance to the post-pandemic world and LGBTQ+ mental health inequalities.
The paper is very well written, crystal clear, and is reporting on a study with a high standard of methodological rigour. It was a pleasure to read. The following comments are intended to strengthen the paper but it should be accepted for publication in this Special Issue subject to these minor corrections.
1. The authors should ensure the Covid-19 pandemic is referred to in past-tense especially in the abstract and introduction.
2. The authors use trauma as an outcome when they describe experiences of discrimination and marginalisation based on identity. Not all people who have these experiences are traumatised. The authors need to think about the claims they make about trauma particularly how this is being conceptualised in relation to prejudice/discrimination etc.
3. In the theoretical framework section, the authors explain they are utilizing a minority coping framework and cite Meyer’s 2015 paper. I think there needs to be further explanations about the minority coping framework and to elaborate on community resilience etc. It may be helpful for the reader to present a figure/diagram/logic model of the authors conceptualisation of minority coping framework.
4. The authors should state how they ensured no harm came to participants given the sample, research topic, and the pandemic environment. What did they do to alleviate stress? What did they have prepared if participants became distressed?
Author Response
The paper is very well written, crystal clear, and is reporting on a study with a high standard of methodological rigour. It was a pleasure to read. The following comments are intended to strengthen the paper but it should be accepted for publication in this Special Issue subject to these minor corrections.
Response: Thank you for your support of our paper and your feedback to help us strengthen its impact.
- The authors should ensure the Covid-19 pandemic is referred to in past-tense especially in the abstract and introduction.
Response: We have carefully revised our paper to ensure that the Covid-19 pandemic is consistently referred to in the past tense throughout.
- The authors use trauma as an outcome when they describe experiences of discrimination and marginalisation based on identity. Not all people who have these experiences are traumatised. The authors need to think about the claims they make about trauma particularly how this is being conceptualised in relation to prejudice/discrimination etc.
Response: We have revised our paper to ensure we are not making sweeping, generalizing comments about how LGBTQ+ people’s experiences of marginalization, including prejudice and discrimination, vary across individuals. We now use more tentative language to state that experiencing marginalization could potentially shape trauma for LGBTQ+ people, such as on page 3: “Identity-based violence can result in potentially traumatic outcomes for marginalized groups.”
- In the theoretical framework section, the authors explain they are utilizing a minority coping framework and cite Meyer’s 2015 paper. I think there needs to be further explanations about the minority coping framework and to elaborate on community resilience etc. It may be helpful for the reader to present a figure/diagram/logic model of the authors conceptualisation of minority coping framework.
Response: We now provide further explanation of how we are utilizing the minority coping framework through additional narrative in-text and the inclusion of Figure 1 on page 5: “For example, minority coping among LGBTQ+ people, including community support systems, self-care practices, and educational empowerment through advocacy, can build resilience in helping LGBTQ+ people overcome minority stressors tied to marginalization [22]”; “We use the minority coping framework to explore LGBTQ+ people’s complex experiences navigating digital geographies amidst a variety of minority stressors online and engaging in minority coping to build resilience during COVID-19 (Figure 1).”
Figure 1. Minority Coping in Online Contexts
- The authors should state how they ensured no harm came to participants given the sample, research topic, and the pandemic environment. What did they do to alleviate stress? What did they have prepared if participants became distressed?
Response: On page 5, we have added more details on how we worked to ensure participant wellbeing: “To ensure participant wellbeing, the interviewers checked in regularly with participants throughout the interview on how they were doing and if they needed a break. Further, each participant was given a list of community-focused resources and the interview team designed a Crisis Protocol to utilize in case a participant exhibited extreme duress, which included providing a mental health hotline and devising a safety plan together (this protocol was never deemed necessary during data collection).”
Reviewer 2 Report
Comments and Suggestions for Authors
Dear Author/s
Overall, this study makes a valuable contribution, but addressing these areas would enhance its impact and depth.
It would be better if the method section was titled as method, procedure, participants, data collection, data analysis.
It should be written more clearly how the semi-structured interview questions were developed
The study touches on implications for health services, social policies, and digital media regulations, but more concrete recommendations would make the findings more actionable.
The study makes an important argument about digital geographies being both empowering and sites of discrimination for LGBTQ+ individuals, but the conclusion should highlight practical takeaways more clearly.
What long-term effects can be expected in the post-pandemic era? Addressing this question in more detail would add depth to the discussion.
Author Response
Overall, this study makes a valuable contribution, but addressing these areas would enhance its impact and depth.
Response: Thank you for your careful review of our paper and your support in helping us strengthen its impact.
It would be better if the method section was titled as method, procedure, participants, data collection, data analysis.
Response: We have added these subheadings as appropriate.
It should be written more clearly how the semi-structured interview questions were developed.
Response: On page 5 we now state: “Specifically, the interview guide questions were developed with a focus on emergent empirical findings surrounding pandemic experiences and through the lens of our guiding theoretical frameworks emphasizing digital geographies and minority stress and coping among LGBTQ+ people.”
The study touches on implications for health services, social policies, and digital media regulations, but more concrete recommendations would make the findings more actionable.
Response: We have added additional concrete recommendations of how our findings could be applied: On page 13: “Furthermore, clinicians can draw from these findings to tailor trauma profiles of clients to be inclusive of vicarious trauma that can occur in digital geographies among marginalized groups”; “Institutions, therefore, should be particularly mindful of their online social settings to ensure inclusive engagement and respectful interactions.” On page 14: “These findings can directly inform community-based support organizations deployment of effective online safe spaces for LGBTQ+ connection.”
The study makes an important argument about digital geographies being both empowering and sites of discrimination for LGBTQ+ individuals, but the conclusion should highlight practical takeaways more clearly.
Response: On page 15, we now state: “Practically, this includes structural recognition of both the power and peril of digital geographies for marginalized groups and implementing organizational practices and policies that center people’s online wellbeing.”
What long-term effects can be expected in the post-pandemic era? Addressing this question in more detail would add depth to the discussion.
Response: We have added some discussion of how our findings can apply to the post-pandemic era: On page 12: “The findings from the early COVID-19 pandemic era suggest long-term implications for LGBTQ+ individuals in the post-pandemic world as the increased reliance on digital spaces has reshaped how LGBTQ+ people navigate identity expression, community building, and activism, creating both opportunities for resilience and challenges like digital burnout and vicarious trauma from online marginalization.” On page 15: “Practically, this includes structural recognition of both the power and peril of digital geographies for marginalized groups and implementing organizational practices and policies that center people’s online wellbeing. LGBTQ+ people’s early pandemic experiences remain relevant in the post-pandemic era by highlighting the need for inclusive policies and practices within digital and institutional spaces to address systemic inequalities, ensure respectful interactions, and support mental health through fostering social connectedness.”